# How Three Self-Secreted Biofilm Exopolysaccharides of *Pseudomonas aeruginosa*, Psl, Pel, and Alginate, Can Each Be Exploited for Antibiotic Adjuvant Effects in Cystic Fibrosis Lung Infection

**DOI:** 10.3390/ijms24108709

**Published:** 2023-05-13

**Authors:** Jonathan Chung, Shafinaz Eisha, Subin Park, Amanda J. Morris, Isaac Martin

**Affiliations:** 1Department of Translational Medicine, Research Institute, The Hospital for Sick Children, University of Toronto, 686 Bay Street, Toronto, ON M5G 0A4, Canada; 2Division of Respiratory Medicine, Department of Paediatrics, The Hospital for Sick Children, University of Toronto, 555 University Avenue, Toronto, ON M5G 1X8, Canada

**Keywords:** cystic fibrosis, *Pseudomonas aeruginosa*, antibiotic adjunct, anti-biofilm, biofilm-degrading enzymes, matrix exopolysaccharides, alginate oligosaccharide

## Abstract

In cystic fibrosis (CF), pulmonary infection with *Pseudomonas aeruginosa* is a cause of increased morbidity and mortality, especially in patients for whom infection becomes chronic and there is reliance on long-term suppressive therapies. Current antimicrobials, though varied mechanistically and by mode of delivery, are inadequate not only due to their failure to eradicate infection but also because they do not halt the progression of lung function decline over time. One of the reasons for this failure is thought to be the biofilm mode of growth of *P. aeruginosa*, wherein self-secreted exopolysaccharides (EPSs) provide physical protection against antibiotics and an array of niches with resulting metabolic and phenotypic heterogeneity. The three biofilm-associated EPSs secreted by *P. aeruginosa* (alginate, Psl, and Pel) are each under investigation and are being exploited in ways that potentiate antibiotics. In this review, we describe the development and structure of *P. aeruginosa* biofilms before examining each EPS as a potential therapeutic target for combating pulmonary infection with *P. aeruginosa* in CF, with a particular focus on the current evidence for these emerging therapies and barriers to bringing these therapies into clinic.

## 1. Introduction

*Pseudomonas aeruginosa* is a ubiquitous, Gram-negative bacterium that occupies environmental niches and is an opportunistic pathogen that poses risks to certain vulnerable groups of patients. Examples include people who are immunocompromised (including those with human immunodeficiency virus or neutropenia stemming from cancer chemotherapy), patients who are post-surgery, burn victims, and individuals with respiratory illnesses, including chronic obstructive pulmonary distress, primary ciliary dyskinesia, and most notably, cystic fibrosis (CF) [1,2,3,4]. CF is a genetic, multisystem disease that results in improper epithelial chloride ion transport. This leads to a dehydrated lung environment, impaired mucociliary escalator function, and the production of sticky mucus that creates suitable grounds for colonization by a number of pathogens, including *P. aeruginosa* [5].

Within the lung, *P. aeruginosa* forms sputum-suspended aggregates (or biofilms) composed of matrix-associated exopolysaccharides (EPSs), proteins, and extracellular DNA (eDNA). This biofilm matrix provides the *P. aeruginosa* biofilm with a scaffolding framework that generates chemical and nutrient gradients that differentially affect cells, giving rise to the incredible metabolic and phenotypic diversity observed through the biofilm [6]. The biofilm matrix also provides physical protection against antimicrobials and host immune cells. In vitro work with antibiotics has consistently demonstrated that biofilm-grown *P. aeruginosa* possess minimum inhibitory concentrations (MICs) that are many orders of magnitude higher than their planktonic counterparts [7,8]. An accumulation of immune-mediated debris from neutrophils and alveolar macrophages results in an aberrant inflammatory response that progressively damages the lung tissue [9]. Managing biofilm infections is an arms race, and as *P. aeruginosa* gradually accumulates adaptations that strengthen its defenses and survivability, our antibiotic arsenal dwindles. In cases where attempts at eradication of this organism fail, the infection becomes chronic, leading to further inflammation and scarring. This damage clinically manifests as progressive lung function decline, decreased quality of life, and increased mortality in CF patients infected with this organism [10,11].

Current strategies employed in clinical practice for eradicating and suppressing *P. aeruginosa* biofilms rely on single or combination high-dose inhaled antibiotic therapy to overcome its naturally high antibiotic tolerance. However, this approach has several drawbacks, including the risk of accumulated side effects (such as ototoxicity or nephrotoxicity with aminoglycoside use) and the emergence of multidrug resistance (MDR) [12,13,14,15]. In 2021, the U.S. CF Foundation reported that among individuals who cultured *P. aeruginosa*, 12.3% were classified as having an MDR strain [16]. This percentage increases with patient age and is thought to represent the cumulative effect of mutagenic adaptation and antibiotic use in the form of chronic suppressive therapy with nebulized antibiotics and courses of antibiotics (nebulized, oral, and intravenous) prescribed in the setting of chronic suppressive therapy as well as acute pulmonary exacerbations. The emergence of MDR not only limits therapeutic options but is associated with worse clinical outcomes in CF [17]. Given the scarcity of new antibiotics, there has been a surge of interest in compounds that may halt the rise of MDR while increasing the efficacy of treatments currently in clinical use (antibiotic adjuncts), some of which target components specific to *P. aeruginosa*. 

Anti-biofilm compounds are an area of interest in CF, and we would direct the reader to a previously published review on this topic [18]. These approaches range from direct bactericidal action to extracellular matrix degradation to disruption of intracellular and cell–cell signaling. In this review, however, we focus on extracellular matrix disruption—more specifically, the three *P. aeruginosa*-produced matrix EPSs—as novel targets for the development of antibiotic adjuncts. We summarize the current evidence base for EPS-degrading enzymes and other EPS-targeting strategies with a focus on their eventual clinical use alongside antibiotics for CF lung infection.

The use of biofilm-degrading compounds to increase the efficacy of antibiotics is an approach with a well-established evidence base in CF. For instance, the use of the mucolytic human recombinant dornase alfa (rhDNase), an agent that degrades eDNA within biofilms, has been shown to be effective at reducing the rate of lung function decline and the number of pulmonary exacerbations in meta-analyses of clinical trials [19]. There are currently no other direct anti-biofilm strategies in clinical use, and, as such, it is difficult to draw direct comparisons. Targeting matrix EPSs specific to the biofilms of *P. aeruginosa* is a promising adjuvant strategy, as these exopolysaccharides play a prominent role in substrate adhesion, cell aggregation, and community architecture, factors that confer survival benefits and lead to increased *P. aeruginosa* tolerance to antimicrobials and host defenses [20,21]. 

## 2. Matrix Exopolysaccharides of *P. aeruginosa* Biofilms

To fully appreciate the potential EPS targets and strategies that will be discussed in this review, we begin with a summary of *P. aeruginosa* biofilm development. The life cycle of the *P. aeruginosa* biofilm is characterized by four stages, depicted in Figure 1: attachment, microcolony formation, biofilm maturation, and dispersal of biofilm bacteria to recolonize and propagate new biofilm development.

The life cycle described in Figure 1 begins when *P. aeruginosa* attaches to a substratum (which may include other cells) and shifts from a highly virulent, motile phenotype to a sessile phenotype. This transition is characterized by the slowing of twitching motility and the secretion of EPSs that anchor bacterial cells [22]. As *P. aeruginosa* proliferates, structured communities called microcolonies form and grow, eventually resulting in the development of mature biofilms. Maturing biofilms are characterized by profound changes in the expression of genes that control twitching motility, EPS secretion, and quorum sensing systems, resulting in an intricately structured, highly responsive, heterogeneous bacterial community [23]. Moreover, genomic studies have revealed that the up- and down-regulation of genes encoding EPSs depends on the bacterial microenvironment [24,25]. When conditions permit, the final stage of the biofilm life cycle is characterized by the dispersal of the mature biofilm, releasing virulent cells and non-surface-attached aggregates to colonize new areas in the CF lung and proliferate infection [26].

The three EPSs of *P. aeruginosa* biofilms include the polysaccharide synthesis locus (Psl), pellicle (Pel), and alginate polysaccharides, each distinct in structure and function within the biofilm, from enabling surface attachment or bacterial aggregation to maintaining architecture and nourishment in the biofilm to offering physical protection from antibiotics and host defenses [27,28]. The specific properties and physiological significance of each EPS in the CF lung will be elaborated upon below.

### 2.1. Psl and Pel

Psl is a neutral, branched pentasaccharide composed of repeating d-mannose, d-glucose, and l-rhamnose subunits and is involved in all stages of *P. aeruginosa* biofilm development [29]. Psl synthesis involves proteins encoded in the *psl* operon, which consists of 15 co-transcribed genes (*pslA-O*) [30,31]. Upon encountering a surface or substrate, Psl, helically arranged on its surface, anchors *P. aeruginosa* and connects it to neighboring cells [22,29,32]. As *P. aeruginosa* swims, it deposits Psl trails that enhance the surface motility of cells, aiding in the formation of microcolonies [33,34]. Psl-sensing further stimulates biofilm development by increasing levels of an important intracellular secondary messenger, cyclic di-guanosine monophosphate [35,36,37]. As the biofilm matures, Psl provides structural integrity by cross-linking CdrA, an extracellular adhesin, and eDNA [22,38,39]. Psl limits the penetration of antibiotics by sequestration and weakens host immunity by preventing efficient complement deposition, thereby inhibiting the production of neutrophil reactive oxygen species (ROS) and efficient phagocytosis of *P. aeruginosa* [40]. Clinically, Psl is associated with increased *P. aeruginosa* tolerance to tobramycin and is associated with aggregation in the sputum of children with CF, which has been put forth as a reason for the failure of tobramycin eradication [41,42]. Although Psl is primarily produced by non-mucoid bacteria, studies have also shown it remains utilized in mucoid CF strains, contributing to biofilm development, maintaining architecture and stability, and evading host immune effectors [43,44,45].

Pel is a cationic polymer important in biofilm formation, specifically for pellicle development at the air-liquid interface, and, such as Psl, Pel is primarily associated with non-mucoid *P. aeruginosa* biofilms [28,46]. Pel is synthesized by proteins expressed by the *pel* operon (*pela-G*) and is composed of galactosamine and N-acetyl galactosamine residues linked by partially de-N-acetylated alpha 1–4 glycosidic linkages [47]. Pel can mediate cell-to-cell interactions through its positive charge, which enables it to cross-link negatively charged eDNA via ionic interactions [48,49]. Furthermore, Pel has been observed to sequester and confer tolerance against positively charged antibiotics such as tobramycin but not neutrally charged antibiotics such as ciprofloxacin [48,49]. Moreover, Pel-eDNA interactions protect eDNA from digestion by DNase I, which, in theory, renders nuclease-based therapies less effective [48,49]. 

Although *P. aeruginosa* can produce both Psl and Pel, their relative contributions to biofilm formation and structural integrity are strain-dependent. In strains that rely on Pel for biofilm formation, such as the wound-derived laboratory strain PA14, Pel acts as a primary scaffold, maintaining cell-to-cell interactions between bacterial cells in a similar fashion as Psl in PAO1 [38,50,51,52]. Indeed, PA14 knockout Pel mutants are unable to form viable biofilms [27]. Likewise, in strains that rely more on Psl, such as the laboratory strain PAO1, knockout Psl mutants inhibit biofilm formation [27]. The localization of Psl and Pel within the biofilm is also strain-dependent. In PAO1, Psl is localized to the periphery of mature microcolonies and bacterial aggregates, while Pel is localized to the base of mature microcolonies [48]. However, in PA14, it is Pel that localizes to the periphery of mature microcolonies [22,48]. The ability to synthesize both Psl and Pel allows for redundancy in which EPS is incorporated into the matrix, highlighting the adaptive capacity of *P. aeruginosa*. For example, it has been observed that Psl-deficient PAO1 mutants upregulate Pel expression in the peripheral region after several repeat cultures [27,48]. Based on these findings, four classes of *P. aeruginosa* strains have been proposed by Colvin et al.: (I) strains that rely on Pel as the dominant matrix polysaccharide (e.g., PA14); (II) strains that rely on Psl as the dominant matrix polysaccharide (e.g., PAO1); (III) strains that produce an EPS redundant matrix where single *psl* or *pel* mutations lead to impaired biofilm formation; and (IV) strains that produce significantly elevated matrix levels such that a single *psl* or *pel* mutation has minimal impact on biofilm biomass [27].

### 2.2. Alginate

Alginate is the most studied and extensively characterized EPS secreted by *P. aeruginosa*. In contrast to Psl and Pel, both of which are specific to *P. aeruginosa*, alginate is utilized by many different bacterial species [53,54,55]. Alginate is an anionic linear polymer with a high molecular weight (MW: 120–480 kDa) and is composed of β-1,4-linked D-mannuronic and α-L-guluronic acids [56,57]. The overproduction of alginate results in a mucoid phenotype and is a significant virulence factor in *P. aeruginosa* CF lung infections [21,58,59]. Clinically, infection with mucoid *P. aeruginosa* is correlated with faster rates of lung function decline compared to non-mucoid strains [60,61,62]. Alginate overproduction is often attributed to an inactivating mutation in the *mucA* gene, which typically encodes an anti-sigma factor that sequesters AlgT, an activator of alginate biosynthesis. Thus, *mucA* mutations result in constitutive alginate synthesis and the appearance of the mucoid phenotype [59,63,64,65].

Although wild-type *P. aeruginosa* has the genetic ability to overproduce alginate, the mucoid phenotype is rarely seen outside the CF lung and arises due to selection pressures in this specific niche. This phenotype has drastic effects on biofilm structure and the persistence of infection within the lung, a fact that is corroborated by the observation that clinical isolates from CF patients often possess alginate-overproducing mucoid phenotypes [65,66,67,68]. Using whole genome sequencing, Marvig et al. followed the evolution of 474 *P. aeruginosa* isolates collected longitudinally from 34 CF patients, beginning at the initial infection and continuing for a mean time span of 4.8 years. They found significant convergent, within-host evolution of many pathoadaptive genes, especially those responsible for alginate overproduction [69]. 

The observed survival advantage and independent development of the mucoid phenotype have been attributed to several factors, specifically the thick and viscous structure of alginate-overproducing biofilms as opposed to the flat and uniform colonies observed in their non-mucoid counterparts [59,70]. In the dehydrated lung mucus of CF patients, alginate helps maintain the hydration of the bacterial cells, a vital function for cell survival and maintenance [65,71]. Furthermore, alginate serves as a protective barrier against antimicrobial therapy and host-mediated immune responses such as macrophage phagocytosis [72,73]. Combined, the alginate exopolysaccharide presents a barrier to antibiotic therapy in CF, and alginate-specific therapeutics will be explored in this review. 

**Figure 1 ijms-24-08709-f001:**
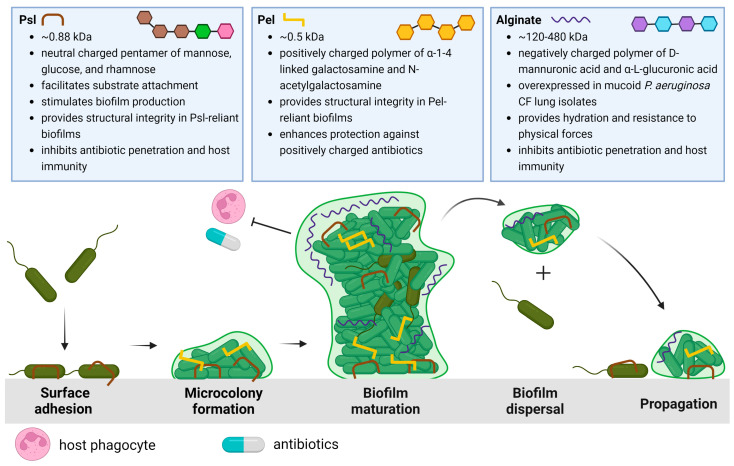
Schematic overview of the *P. aeruginosa* life cycle and the characteristics and roles of matrix exopolysaccharides Psl [22,29,32,36,38,39,40], Pel [47,48,49,52], and alginate [56,58,59,65,71,72,73]. The biofilm life cycle begins with Psl-mediated surface/substrate attachment of *P. aeruginosa*. The association of *P. aeruginosa* cells results in microcolony formation as motile cells transition to a sessile phenotype. As the biofilm develops, Psl, Pel, or both EPSs are incorporated into the biofilm architecture to form large 3D macrostructures. The mature biofilm is characterized by the differential expression of these EPSs, the activation of quorum sensing, increasing population heterogeneity, and eventual dispersal to new areas of the lung. Created with Biorender.com.

## 3. Therapies

Proposed anti-biofilm therapies employing matrix-EPS degrading enzymes include PslG glycoside hydrolase (PslG_h_), PelA glycoside hydrolase (PelA_h_), and alginate lyase (AgLase). While PslG_h_ and PelA_h_ are both synthesized by and specific to *P. aeruginosa* matrix EPSs, AgLase can be isolated from a variety of sources, including algae, mollusks, bacteria, and viruses. Another approach has been with the alginate oligomer “OligoG CF-5/20”, which alters the viscoelastic properties of mucus through mucin-alginate interactions in CF sputum [74]. Other therapies that target *P. aeruginosa* EPS, including non-specific glycoside hydrolases, bacteriophage, and monoclonal antibodies, will also be briefly touched upon but are not the focus of this review. The known properties, effects, and toxicities of these compounds are summarized in Figure 2.

### 3.1. Specific EPS Therapies

#### 3.1.1. Matrix Degrading Enzymes

PslG is a 47.38 kDa molecular weight (MW) periplasmic protein involved in Psl synthesis and is co-transcribed with 15 other genes in the *psl* operon [31,75]. PslG contains an electronegative glycoside hydrolase domain (PslG_h_) that degrades Psl chains by hydrolyzing adjacent mannose residues [31,75]. PelA is a 101.3 kDa multidomain, periplasmic protein encoded by the first gene of the *pel* operon and contains a deacetylase domain and a glycoside hydrolase domain with distinct functions [27,76]. Its deacetylase domain bestows Pel with a positive charge, allowing it to associate with other charged elements in the biofilm matrix such as eDNA [48,76,77]. Its glycoside hydrolase domain (MW: 30.58 kDa), herein PelA_h_, hydrolyzes cationic, partially deacetylated residues within Pel residues and is involved in biofilm dispersal [78,79,80]. Given the highly selective action of PslG_h_ and PelA_h_ against their substrates, Psl and Pel, which have critical roles in biofilm development and functioning, PslG_h_ and PelA_h_ have been studied as adjuvant therapies for eradicating *P. aeruginosa* biofilms.

Several lines of evidence from in vitro studies support the potential of PslG_h_ and PelA_h_ for the treatment of *P. aeruginosa* biofilms. When supplied exogenously, PslG_h_ and PelA_h_ can disrupt mature *P. aeruginosa* biofilms at nanomolar concentrations with extremely high specificity [75,79]. In developing biofilms, PslG_h_ impairs the ability of *P. aeruginosa* to associate with surfaces and other bacteria, leading to more rapid, scattered movement of *P. aeruginosa*, delaying microcolony formation, and inhibiting biofilm development [81]. When combined, PslG_h-_PelA_h_ improved the penetration and efficacy of ciprofloxacin, tobramycin, colistin, neomycin, and polymyxin, permitting antibiotic dose reduction [82,83]. PslG_h_ also works remarkably quickly, disrupting Psl fibers within 5 min of administration and triggering sudden biofilm dispersal [75]. Moreover, PslG_h_-dispersed bacteria possess lower MIC values for tobramycin and ciprofloxacin compared to planktonic bacteria [75]. PslG_h_ further sensitizes the host immune system towards *P. aeruginosa* biofilms, increasing the deposition of C3 complement proteins, stimulating macrophage and neutrophil phagocytosis, and enhancing neutrophil ROS production [31,75,83]. When tested in vitro, both PslG_h_ and PelA_h_ were observed to be non-toxic to neutrophils, lung fibroblasts, and red blood cells, and PslG_h_ was additionally non-toxic to colonic epithelial cells and macrophages [31,75,83]. Experiments on animal models of *P. aeruginosa* have validated in vitro findings regarding the efficacy and safety of PslG_h_ and PelA_h_. When injected locally, PslG_h_ potentiated tobramycin’s killing of mature *P. aeruginosa* biofilms on mouse peritoneum implants [75]. In a wound infection model, prophylactic treatment with PslG_h_ showed an additive killing effect when tobramycin was added 24 h later and was not toxic to the wound [83]. PelA_h_ embedded in bacterial cellulose membranes effectively destabilized *P. aeruginosa* biofilms when topically applied to infected murine chronic wounds [84]. Although the majority of studies have utilized wound and implant models, these results are promising and demonstrate the proof of concept for topical (i.e., nebulized) application in pulmonary infection in CF.

In the most comprehensive pre-clinical evaluation of PslG_h_ and PelA_h_ to date, Ostapska et al. prophylactically administered PslG_h_ and PelA_h_ intratracheally to mice with *P. aeruginosa* lung infections with or without antibiotics [82]. They found that the antimicrobial effects of ciprofloxacin, but not ceftazidime, were potentiated when administered every 8 h following initial treatment with PslG_h_-PelA_h_. In healthy mice, a single intratracheal dose of up to 250/250 µg of PslG_h_-PelA_h_ was well tolerated, with no changes in weight, temperature, mortality, markers of pulmonary injury, or numbers of macrophage, eosinophil, and neutrophil cells compared to buffer-treated mice. However, there was an increase in pulmonary lymphocytes following PslG_h_-PelA_h_ treatment, highlighting the need to further investigate the adaptive immune response to this therapy. When administered alone, PslG_h_ has a half-life of 18 h, and PelA_h_ has a half-life of 3 h. However, PelA_h_ stability increased to 5 h when co-administered with PslG_h_, and its catalytic activity was retained for >24 h. It is worth noting, however, that PslG_h_-PelA_h_ treatment of *P. aeruginosa*-infected mice in the absence of antibiotics triggered pulmonary inflammation and lethal septicemia, underscoring the necessity of antibiotic co-treatment [82].

Several technological developments that may enhance the feasibility of PslG_h_ and PelA_h_ in clinical applications are underway. Lipid liquid crystal nanoparticles encapsulating PslG_h_ and tobramycin offer protection against proteolytic degradation and only release antibiotic contents upon encountering *P. aeruginosa*. This product was seen to be 10–100 fold more effective at eradicating *P. aeruginosa* infections in vivo using a *Caenorhabditis elegans* infection model [85]. PslG_h_ constructs with significantly enhanced trypsin resistance have also been developed, which may extend the half-life of PslG_h_, allowing for smaller and less frequent doses [86]. PslG_h_ can also be immobilized on the lumen surface of medical-grade polyethylene, polyurethane, and polydimethylsiloxane (silicone) catheter tubing, reducing the surface attachment of *P. aeruginosa*. PslG_h_-modified catheters decreased the bacterial burden of *P. aeruginosa* by 3 logs at 11 days under dynamic flow culture conditions in vitro and decreased the burden of *P. aeruginosa* by 1.5 logs in an in vivo rat infection model [87].

Published research has reported that mucoid, alginate-rich *P. aeruginosa* isolates produce heterogeneous biofilms with high tolerance to antibiotics [59]. Many anti-biofilm strategies have been designed and examined for increasing the susceptibility of antibiotics through disrupting the alginate EPS. One such promising therapeutic technique is using the enzyme AgLase, which can be derived from various sources including algae, mollusks, bacteria, viruses, and fungi and weighs between 25 and 60 kDa [88]. AgLase degrades alginate at a variety of cleavage points depending on its source, thereby disrupting mucoid *P. aeruginosa* biofilms and enhancing antibiotic penetration and host immune functions [88]. AgLase has been reported in vitro and in vivo to disrupt biofilms, enhancing the efficacy of amikacin, tobramycin, ciprofloxacin, and gentamicin [72,89,90]. AgLase co-administration with DNase has been found to potentiate antimicrobial biofilm eradication to a greater effect than seen with either agent on its own [91,92]. AgLase has also been seen to enhance neutrophil killing, macrophage phagocytosis, and alveolar macrophage efferocytosis [73,93,94]. There is, however, conflicting evidence for AgLase as an antibiotic adjunct. One study reported that AgLase did not have any effect on pure or mixed cultures of *P. aeruginosa* biofilms, a fact that the authors attributed in certain instances to alginate not being the main contributing component of the biofilm and in others to protection from enzymatic degradation due to the presence of other molecules [95]. Another study compared two AgLases to proteins similar in structure but without the capacity to enzymatically degrade alginate and observed equal rates of bacterial biofilm disruption and antibiotic synergy between these compounds, suggesting that the potentiation effects of AgLases are uncoupled from their catalytic activity [96]. Another in vitro study found that the AgLase used in their experiments was also unable to degrade mucoid *P. aeruginosa* biofilms embedded in sputum [97].

Though the evidence is unclear as to whether and how AgLases combat bacterial biofilms, AgLase products with therapeutic implications are under development. AgLase-polyethylene glycol conjugates have been developed that significantly reduce its immunoreactivity [98]. AgLase-functionalized chitosan nanoparticles of ciprofloxacin are novel AgLase delivery systems that have demonstrated enhanced biofilm degradation in in vitro experiments with a mucoid clinical *P. aeruginosa* strain without toxicity to human lung epithelial cells or rat lung tissues [99]. 

#### 3.1.2. OligoG

Alongside the development of matrix-degrading enzymes as adjunctive therapies for CF patients, other novel therapeutics that may enhance airway clearance and lung function are being investigated. One such example is the 3.2 kDa alginate oligosaccharide, OligoG CF-5/20 (OligoG), which is sourced from the brown seaweed *Laminaria hyperborean* and can reduce the viscoelasticity of CF sputum [57,74,100]. These low molecular weight oligosaccharides displace mucin-high molecular weight alginate interactions, readily diffuse through mucoid biofilms, and disrupt EPS within biofilm matrices [57,101]. OligoG disrupts biofilms in a time- and dose-dependent manner and can act as an antibiotic potentiator, increasing the penetration and efficacy of tobramycin, erythromycin, colistin, and ciprofloxacin [57,101,102,103]. Mice intratracheally infected with a mucoid clinical *P. aeruginosa* isolate showed a 2.5 log reduction in bacterial colony-forming units (CFUs) following treatment with 5% OligoG [102]. As with the aforementioned matrix disruptors, OligoG is not bactericidal; however, OligoG-colistin conjugates have been developed that would theoretically allow for lower effective dosing, thereby limiting the toxic side effects of this antibiotic [104]. There are also novel compounds under development that combine the mucoactive properties of alginate oligomers with nitric oxide release as an added antibiofilm agent to improve the efficacy of antibiotics [105].

Importantly, OligoG is the first inhaled polymer therapy that has been investigated in humans as a novel therapeutic approach to aid airway clearance in CF patients. AlgiPharma, the lead developer of OligoG, has been spearheading efforts to develop OligoG as a novel CF therapeutic and has completed several clinical trials, although the results of only one Phase 2b trial have been published so far. In this randomized, double-blind, placebo-controlled, multi-center crossover study, 65 adult CF patients were randomly allocated to receive either OligoG via dry powder inhalation (1050 mg per day) or a placebo. The primary endpoint was forced expiratory volume in 1 s (FEV1), measured at the end of the 28-day treatment regimen [106]. This study revealed that OligoG administered thrice daily via dry powder inhalation was well tolerated, with no significant differences in serious adverse events between the treatment and placebo groups. Furthermore, the OligoG concentration in blood plasma was in the range of 0.5–8.98 µg∙mL^−1^, with no detectable plasma OligoG after 28 days of washout (day 56 of the study). Despite this reassuring safety data, however, no significant improvement in FEV1, the primary endpoint, was observed in the treatment group. Interestingly, post hoc exploratory analyses indicated that patients on inhaled tobramycin and patients <25 years of age showed positive trends in lung function, highlighting that further studies of OligoG are warranted [106]. A prospective clinical study is currently being planned under the framework of HORIZON2020 by the European Clinical Trial Network [107]. Whatever the outcome, these trials will shed new light on the therapeutic potential of disrupting alginate biofilms in *P. aeruginosa* pulmonary infection.

#### 3.1.3. Monoclonal Antibodies

In the realm of immunotherapy, monoclonal antibodies (mAbs) are a novel approach to targeting *P. aeruginosa* lung infections in CF. As *P. aeruginosa* is known to evade immune responses through a combination of factors, including Psl, passive immunization with monoclonal antibodies to improve host immunity is a promising therapeutic strategy [40,108]. Acting non-enzymatically, anti-psl and anti-alginate mAbs can aid recognition of *P. aeruginosa* by the immune system and have indeed been shown to increase opsonic and complement-mediated immune killing of *P. aeruginosa* both in vitro and in animal models [108,109,110,111]. These mAbs have further been shown to reduce cell-surface attachment, invasion, and cell aggregation, which are crucial functions for the initiation and virulence of *P. aeruginosa* biofilms [108,111]. Although other monoclonal antibodies targeting different antigens of *P. aeruginosa* have been developed, they will not be discussed in this review. 

Given early promising in vivo data, two clinical anti-Psl and anti-alginate mAb candidates were developed: MEDI3902 (200 kDa), a multi-mechanistic bispecific antibody targeting Psl, and AR-105, a monoclonal antibody that targets alginate [112,113]. MEDI3902 was effective in vitro against a variety of *P. aeruginosa* strains, synergizing with multiple antibiotic classes and imparting prophylactic protection against *P. aeruginosa* in various mice infection models [113]. Prophylaxis with MED3902 in a rabbit *P. aeruginosa* acute pneumonia model was protective and further decreased proinflammatory mediators and lung tissue damage [114]. However, despite reassuring safety data in Phase 1 clinical trials, neither MEDI3902 nor AR-105 satisfied the primary endpoint in Phase 2 clinical trials involving patients with pneumonia caused by *P. aeruginosa* (NCT02696902) or (NCT03027609) [115,116,117,118]. 

### 3.2. Non-Specific EPS Disruptors

#### 3.2.1. Non-Specific Glycoside Hydrolases

Another strategy for inhibition and removal of biofilm is disrupting exopolysaccharides via the application of non-specific glycoside hydrolases. In vitro studies have shown that α-amylase, a glycoside hydrolase isolated from Bacillus subtilis, possesses the ability to inhibit and eradicate biofilms of two bacterial species, *S. aureus* and *P. aeruginosa* [119,120,121]. The crude enzyme was effective against *S. aureus* and *P. aeruginosa*, inhibiting 51.8% to 73.1% of biofilm formation while also degrading mature biofilms by disrupting exopolysaccharides [120]. There are other amylases with biofilm-degrading activity, such as β-amylase, but it does not work as quickly as α-amylase [119,122].

Cellulase is a glycoside hydrolase enzyme observed to be partially effective in reducing biomass and decreasing colony-forming units of *P. aeruginosa* [123]. The experiments conducted by Loiselle and Anderson revealed a pH- and dose-dependent biofilm degradation effect of cellulase [123]. The decrease in molecular weight of EPS fractions incubated with cellulase suggests the ability of cellulase to degrade *P. aeruginosa* EPS [123]. When cellulase was combined with the antibiotic ceftazidime, there was a synergistic inhibition and eradication of *P. aeruginosa* biofilm [124]. Since cellulase does not completely inhibit or eradicate the biofilm, this enzyme can be used in combination with other treatments to increase the efficiency of biofilm degradation or lower antibiotic dosages to safer levels. 

Other hydrolases, such as α-mannosidase and β-mannosidase were reported to be effective in the degradation of *P. aeruginosa* biofilms by degrading Psl, which is rich in mannose residues [125]. However, mannosidases were found to be cytotoxic to human cell lines by reducing mitochondrial activity and altering cell morphology, disincentivizing further clinical development [125]. Moreover, lyticase and β-glucosidase are non-cytotoxic enzymes that disrupt the Psl and Pel polysaccharides, degrading the *P. aeruginosa* biofilm and reducing colony forming units [126].

#### 3.2.2. Bacteriophage

Lytic bacteriophages are viruses that lyse bacterial cells by attaching to bacterial surface receptors and injecting their genomic DNA/RNA, using the host cellular machinery to replicate and eventually lyse the cell. It is known that some antipseudomonal bacteriophages employ polysaccharide depolymerases to burrow through polysaccharide-rich biofilms before infecting the biofilm cells embedded within [127,128]. One study revealed that *P. aeruginosa*-specific bacteriophage could reduce the viscosity of purified CF alginate by up to 40% and penetrate through the mucoid alginate matrix to lyse biofilm-associated bacteria [129]. Glonti et al. isolated an alginate lyase responsible for the alginate-degrading properties of a *P. aeruginosa* bacteriophage and reported activity against alginate purified from many sources, including clinical CF isolates [130]. Similarly, Alemayehu et al. isolated two phages from a water treatment plant and demonstrated that an equal mixture of the two phages had lethal effects on *P. aeruginosa* in human lung epithelial cells from CF patients and in murine models [131]. A *P. aeruginosa* phage isolated from hospital sewage encoding a glycoside hydrolase was able to potentiate bactericidal activity against and disrupt mature *P. aeruginosa* biofilms [132]. A number of clinical trials involving the use of bacteriophages for *P. aeruginosa* lung infection in CF are currently underway, and although the contribution of specific depolymerases to the antibiofilm properties of *P. aeruginosa* bacteriophages is yet to be determined, it will be interesting to see whether the presence of such enzymes will play a role in clinical response (NCT04596319; NCT04684641; NCT05453578; NCT05010577) [133,134,135,136]. 

**Figure 2 ijms-24-08709-f002:**
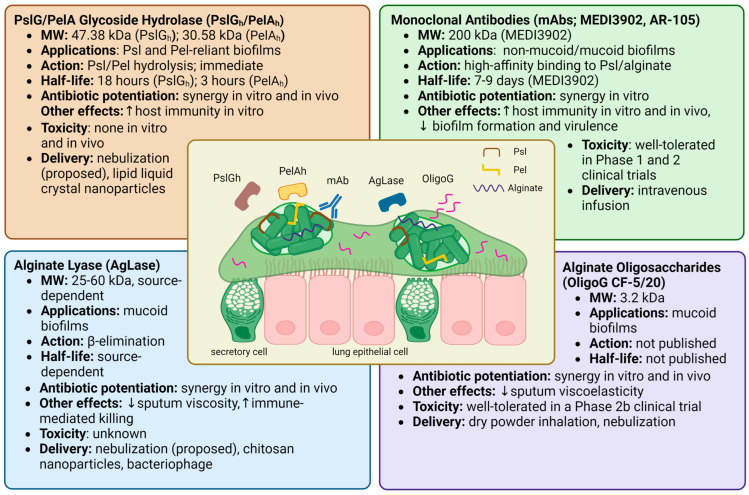
Overview of the therapeutics known to target specific *P. aeruginosa* biofilm EPSs. The central illustration shows bacterial aggregates suspended in CF airway sputum. Outlined are potential therapeutic applications, drug mechanisms, half-life, antibiotic synergy, immune effects, and safety data for PslG/PelA glycoside hydrolase [29,75,78,79,80,82,83,85], alginate lyase [72,88,89,90,93,94,99,129,133,134,135,136], monoclonal antibodies [108,109,110,111,112,113,115,116,117,118], and alginate oligosaccharides [58,74,101,102,103,106]. Created with Biorender.com.

## 4. Therapeutic Challenges and Considerations

Successful treatment of *P. aeruginosa* CF pulmonary infections is difficult to achieve, in part due to the heterogeneity of bacterial populations within the biofilm (cells with differing metabolism, mutation rate, phenotypic expression of virulence factors, etc.) that limits the effectiveness of broad-spectrum antibiotics at our disposal. The agents described in this review all have the potential to disrupt the biofilm structure. Although not bactericidal by themselves, the hope is that an adjuvant effect will result from disrupting the extracellular matrix, exposing bacterial cells to both effectors of the immune system (such as alveolar macrophages and neutrophils) as well as antibiotics. Additionally, their selectivity for targeting *P. aeruginosa* infections, except for the non-specific glycoside hydrolases, makes them appealing adjuncts in an era of personalized medicine while limiting off-target effects. These agents are summarized in Table 1, which focuses on the challenges and considerations that exist as these compounds transition from laboratory bench to CF patient or from clinical trial to effective therapy.

One should note that some of the compounds mentioned in Table 1 are further along the development pipeline than others, with OligoG, bacteriophage, and monoclonal antibodies being the only compounds being administered to humans in clinical trials. Although the OligoG trial failed to meet its primary endpoint (improvement in FEV1), there were several trends in post hoc analyses that should be taken into consideration for clinical trials in development, including the use of a placebo inhaler that unexpectedly reduced bacterial burden in sputum and some promising trends in certain patient groups [106]. Furthermore, although the Phase 2 monoclonal antibody trials in *P. aeruginosa* pneumonia also did not reach their primary endpoints, they demonstrated acceptable Phase 1 safety profiles and may prove beneficial to different patient groups, including CF patients with chronic rather than acute infections.

The matrix-degrading enzymes PslG_h_, PelA_h_, and AgLase have all shown efficacy in vitro and in vivo as potent biofilm disruptors and antibiotic potentiators, although it is clear (Table 1) that if these compounds are to be developed into viable antibiotic adjuncts in CF patients, more research is needed. Specifically, investigations across many different clinical strains are required to demonstrate whether there is the potential for widespread applicability or whether the promising effects that have been observed are specific only to certain bacterial strains. To overcome this high level of specificity, studies bringing these therapies to animal models have focused on co-administration of other compounds, whether it is hrDNase or a combination of the glycoside hydrolases mentioned in this review [97,124]. Combining different matrix-degrading enzymes may have many benefits, from targeting multiple EPSs at once and achieving broader therapeutic coverage to extending the half-life of the compounds, as was observed when PelA_h_ was combined with PslG_h_ in previous experiments [82]. 

Moreover, there is no reason that antibiotics with different mechanisms of action should all act in additive or synergistic ways with these compounds, and this is illustrated by some of the preliminary in vitro work showing additional benefit when PslG_h_ was administered with the fluoroquinolone ciprofloxacin but not with the third-generation cephalosporin ceftazidime—both commonly used agents in *P. aeruginosa* infection [82]. It would be worthwhile to know whether specific antibiotic combinations have more synergistic potential than others if and when these agents are developed for human use and brought to clinical trials, as these findings might inform analyses of patient subgroups on different antibiotics. 

Additional information is also needed for finding optimal dose concentrations for both safety and efficacy. Although PslG_h_ and PelA_h_ have been demonstrated to be non-cytotoxic to mammalian cells in all the studies included in this review, Ostapska et al. showed an increase in the number of pulmonary lymphocytes in mice following glycoside hydrolase administration, warranting further study into adaptive immune responses with repeated exposure. Of critical importance are the hematological dissemination and lethal septicemia that have been seen in animal models if glycoside hydrolases are used in the absence of antibiotics [82,137]. Furthermore, many of the in vivo studies looking into the antibiotic potentiation effects of these compounds use wound or implant infection models, leaving a gap in knowledge concerning the efficacy of these compounds in pulmonary infection models that better represent the lung environment in CF. 

Concerning the mode of administration for these compounds, except for OligoG, it is unknown whether (i) they are stable when nebulized or inhaled and (ii) they can be administered in sufficient quantity to exert clinically significant antibiotic potentiation. Given that the glycoside hydrolases PslG_h_ and PelA_h_ disperse biofilms almost immediately, it seems reasonable to develop these compounds either for administration prior to or for co-administration with antibiotics. Furthermore, there is a lack of information on the stability of these compounds in CF sputum, where it is possible that the presence of proteases and other specifics of the environmental niche (e.g., temperature, pH, etc.) could denature these enzymes before they can degrade the biofilm. Some of these concerns may be addressed by technological modifications briefly introduced in this review, such as novel lipid liquid crystal and chitosan nanoparticles to enhance stability and delivery, protease-resistant hydrolases/lyases to sustain duration of action, and polyethylene glycol conjugation to decrease immunoreactivity.

Lastly, we must address the chasm that exists between in vitro laboratory methods for testing antibiotic sensitivity and clinical outcomes in CF lung infections [140]. This is likely due to the simplicity of in vitro assays, their inadequacy in simulating a biofilm mode of growth, and the myriad of conditions encountered in the human lung. However, even when clinical trials have assessed the choice of antibiotic based on assays that have used bacterial biofilms, there has been no link with patient outcome [141]. There are a variety of assays that have been developed/are in development that aim to rectify this disconnect, including the use of artificial or CF sputum, animal models of pulmonary infection, and infection models that explore the polymicrobial nature of the lung.

Concerning the latter, *P. aeruginosa* matrix EPSs have also been noted to play important roles in mixed-species communities, conferring both communal benefits and competitive advantages. For instance, Psl has been noted to integrate *P. aeruginosa* into mixed-species biofilms with *Pseudomonas protegens* and *Klebsiella pneumoniae* and confer communal stress resistance [142]. Both Psl and Pel have been implicated in the development of mixed-species biofilms with *Staphylococcus aureus* and were seen to promote antibiotic tolerance in Psl non-producers such as *Escherichia coli* and *Staphylococcus aureus* by binding secreted staphylococcal protein A [143,144,145]. In other studies looking at mucoid *P. aeruginosa* isolates from the CF lung, the overproduction of alginate not only rendered growth conditions more permissive to the growth of *S. aureus* [146], but also protected *Burkholderia cenocepacia* from inflammatory mediators and host immune effectors [147]. Thus, it is possible that a disruption of *P. aeruginosa* matrix EPSs may yield a therapeutic effect beyond the scope of the primary infection [148].

## 5. Conclusions

Despite tremendous progress in our understanding of bacterial biofilms in recent years, *P. aeruginosa* lung infection remains a significant cause of morbidity and mortality in the CF population [11]. The recent introduction of highly effective modulator therapy (HEMT) in CF has shown great promise in restoring some CFTR protein function and thus impacting mucus and the CF lung environment in clinically meaningful ways, improving pulmonary function, BMI, patient-reported quality of life, and decreasing rates of pulmonary exacerbations [149]. Additionally, there have been reports that the modulator ivacaftor (used in most of the currently available HEMT) might possess some antimicrobial activity both directly [150] and when combined with common antibiotics [151,152]. Though it is not yet clear what impact these therapies will have on lung infection, by “normalizing” the properties of pulmonary mucus, the hope is that younger patients starting on these therapies will not be affected by common CF pathogens. However, current evidence suggests that even with HEMT, CF lung disease still progresses, albeit slower, and organisms such as *P. aeruginosa* will continue to be problematic for the majority of individuals with well-established lung disease [153,154,155]. Additionally, the high cost of HEMT and lack of uniform provision across different health systems mean that CF patients in developing countries will not be able to benefit from these compounds in the short term [149]. Additionally, there is a population of CF patients with CFTR mutations not amenable to treatment by current HEMT and others that cannot tolerate these medications due to adverse hepatic, somatic, or psychiatric symptoms [153]. It is thus important to continue searching for novel antimicrobial strategies and compounds to improve outcomes for this patient population.

The three EPSs produced by *P. aeruginosa* (Psl, Pel, and alginate) each represent novel targets for antibiotic adjuvant therapy. As we enter an era of more personalized medicine, tailored therapies may hold the key to antibiotic potentiation that we have been searching for, whether the target is an alginate over-producing mucus isolate or a Psl/Pel-rich one. Despite this promise, the only compounds discussed in this review that have been given in clinical trials (OligoG, MEDI3902, and AR-105) failed to meet the primary endpoints at the end of the trial period. Many of the other compounds, including matrix-degrading enzymes, have only been administered in animals, and data are lacking on their applicability across clinical strains found in the CF lung. Hence, there are several steps before such compounds come to human clinical trials. Further in vitro work will be needed to establish the efficacy of these compounds across a variety of clinical strains, in addition to considering other clinically relevant questions, including optimizing concentrations and assessing the mode and timing of delivery in relation to other therapies.

## Figures and Tables

**Table 1 ijms-24-08709-t001:** Therapies exploiting *P. aeruginosa* biofilm exopolysaccharides.

Product	Therapeutic Challenges and Considerations
**Specific EPS Therapies**	
**PslG glycoside hydrolase (PslG_h_)** **+/−** **PelA glycoside hydrolase (PelA_h_)**	Unclear relevance of both Psl and Pel to biofilm infections across clinical isolates; Uncertain stability of both compounds in CF sputum; Optimal dosing concentration and regimen unknown; Risk of disseminated infection if not given with antimicrobials [82].
**Alginate Lyase (AgLase)**	Unclear stability of compounds in CF sputum; AgLases isolated from diverse sources have distinct enzymatic profiles [88]; Data conflicting on antibiotic potentiation effects (e.g., not as effective when alginate not main component of biofilm) [95]; Biofilm degrading capacity of AgLase may be independent of its catalytic activity [96]; AgLase degradation of alginate in CF mucoid *P. aeruginosa* biofilm was ineffective in vitro, when incubated with sputum [97].
**Alginate Oligosaccharide** **(OligoG, CF-5/20)**	Primary endpoint (improved FEV1) not met in Phase 2b clinical trial [106]; A total of 45% of patients did not adhere to three times daily dosing regimen; Younger patients and those on continuous or inhaled antibiotics showed trend towards improved efficacy; Larger Phase 3 clinical trials needed.
**Anti-EPS Monoclonal Antibodies (MEDI3902, AR-105)**	Unsuccessful Phase 2 clinical trials in *P. aeruginosa* pneumonia patients; but no clinical trials have been conducted with CF patients; Possible interference between monoclonal antibodies and host’s own immune defense [115]; Uncertainty towards in achieving adequate monoclonal antibody coverage in lung epithelial lining [115].
**Non-Specific EPS Therapies**	
**Non-specific glycoside hydrolases**	Unclear stability of compounds in CF sputum; Broad enzyme activity may cause off-target effects; Optimal dosing concentration and delivery system unknown; Risk of disseminated infection if not given with antimicrobials [137].
**Bacteriophages**	Utilize a variety of mechanisms including direct bactericidal action, but may also encode for and utilize various EPS depolymerases [129,130,131,132]; Antipseudomonal phages have been well tolerated when used on a compassionate basis in CF patients in both nebulized and intravenous formulations [138,139]; Current clinical trials will examine safety and tolerability, but also efficacy as assessed by outcome measures such as FEV1 [133,134,135,136].

## Data Availability

Not applicable.

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
