# Peer review of "How Three Self-Secreted Biofilm Exopolysaccharides of Pseudomonas aeruginosa, Psl, Pel, and Alginate, Can Each Be Exploited for Antibiotic Adjuvant Effects in Cystic Fibrosis Lung Infection"

_ijms, 2023, doi:10.3390/ijms24108709_

Round 1

Reviewer 1 Report

This is a well written review describing the well known issues with biofilms in CF patients infection with p auruginosa. There is sufficient detail presented for the reader to understand the basic biochemical mechanisms related to biofilms and the problems treating pulmonary infections accompanied by such biofilms. A particular focus on tx discovery and development is given. The reviewer finds little to criticize and the review should be of interest to those in medicin as well as those with interest in antibacterial discovery and development. The review deserves publication after editorial review for potential minor spelling, punctiation issues.

Author Response

We thank the reviewer for reading the manuscript and for the encouraging feedback. 

Please feel free to read the revised manuscript where we have improved punctuation and grammar, and in certain cases have improved flow. 

There are also some significant changes made to the "Therapeutic Challenges and Considerations" section. 

Many thanks,

Authors

Reviewer 2 Report

The manuscript, entitled “Exploiting biofilm-matrix exopolysaccharides with antibiotic adjuvants in the treatment of Pseudomonas aeruginosa in cystic fibrosis lung infection” focused on the disruption of the three P. aeruginosa-specific matrix EPSs as novel therapeutic targets for increasing the efficacy of antibiotics currently in clinical use and summarized the current evidence base for EPS degrading enzymes and other strategies with a focus on the eventual clinical use alongside antibiotics for CF lung infection. This is a meaningful information; my suggestion is major revision.

Comments:

1. If there are only three different exopolysaccharides including Psl, Pel, Alginate in biofilm matrix of Pseudomonas aeruginosa? Why did the authors focus on these three kinds of exopolysaccharides? What’s more, what is the novelties or advantages of this manuscript?

2. The efficiency of exopolysaccharides as antibiotic adjuvants compared with other therapies is not mentioned. It is important information for this review.

3. The content on “Therapeutic Challenges and Considerations” is not comprehensive and perfect enough. Please supplement more and detailed information.

4. What’s the full name of “Psl” and “Pel”? The first appearance of the abbreviation should be attached with it’s full name.

5. The format of “EPS” is inconsistent. In line 84 EPS is in italics.

6. From line 90 to line 97, in the figure legend of Figure 1, “P. aeruginosa” should be in italics.

7. In line 193, the cite format of references is not consistent with others.

Author Response

Dear reviewer, 

Thank you for your thoughtful comments. Please see the attachment for how we have addressed each concern. 

Round 2

Reviewer 2 Report

The authors revised the manuscript partly, however, due to some drawbacks left, my suggestion is rejection.

Major Comments:

1. First and foremost, this manuscript focused on that exopolysaccharides including Psl, Pel, and alginate can be used as antibiotics adjuvant to treat cystic fibrosis caused by P. aeruginosa. The part followed introduction is about these three exopolysaccharides, then therapies are shown. However, in the section of therapies section, two sections “3.1. Matrix degrading enzymes” and “3.2. Other novel strategies” are included. In section of 3.2, only OligoG was introduced mainly. Whats more the mechanisms of these therapies were not explained clearly, and the authors just listed some papaers published. The paper published in 2014 (https://doi.org/10.1016/j.biologicals.2013.11.001) concluded more therapies. So, the innovation and comprehensiveness of this review are lacking.

2. For the information in section “4. Therapeutic Challenges and Considerations”, it is suggested to conclude the information in a figure or table.

3. Among the references cited, fewer articles have been published in the last five years. The manuscript does not cover the relevant content of recent years well.

4. The title is inappropriate and needs to be further refined.

Minor Comments:

1. In line 84, there should be a blank space between EPS and targets.

2. In line 84, it shows to appreciate the potential EPS, but this review mainly focused on exopolysaccharides.

3. The authors provide some basic information about psl, pel, and alginate, but more detailed function of these three exopolysaccharides on anchoring to cells was not given. For example, what is the main

4. In line 171, it should be utilized by.

5. In line 175, the format of lung infection. [20,57,58] is wrong.
